# Reflection methods for user-friendly submodular optimization

**Stefanie Jegelka**
UC Berkeley
Berkeley, CA, USA

**Francis Bach**
INRIA - ENS
Paris, France

**Suvrit Sra**
MPI for Intelligent Systems
Tübingen, Germany

## Abstract

Recently, it has become evident that submodularity naturally captures widely occurring concepts in machine learning, signal processing and computer vision. Consequently, there is need for efficient optimization procedures for submodular functions, especially for minimization problems. While general submodular minimization is challenging, we propose a new method that exploits existing decomposability of submodular functions. In contrast to previous approaches, our method is neither approximate, nor impractical, nor does it need any cumbersome parameter tuning. Moreover, it is easy to implement and parallelize. A key component of our method is a formulation of the discrete submodular minimization problem as a continuous best approximation problem that is solved through a sequence of reflections, and its solution can be easily thresholded to obtain an optimal discrete solution. This method solves *both* the continuous and discrete formulations of the problem, and therefore has applications in learning, inference, and reconstruction. In our experiments, we illustrate the benefits of our method on two image segmentation tasks.

## 1 Introduction

Submodularity is a rich combinatorial concept that expresses widely occurring phenomena such as diminishing marginal costs and preferences for grouping. A set function $F : 2^V \to \mathbb{R}$ on a set $V$ is *submodular* if for all subsets $S, T \subseteq V$, we have $F(S \cup T) + F(S \cap T) \leq F(S) + F(T)$.

Submodular functions underlie the goals of numerous problems in machine learning, computer vision and signal processing [1]. Several problems in these areas can be phrased as submodular optimization tasks: notable examples include graph cut-based image segmentation [7], sensor placement [30], or document summarization [31]. A longer list of examples may be found in [1].

The theoretical complexity of submodular optimization is well-understood: unconstrained minimization of submodular set functions is polynomial-time [19] while submodular maximization is NP-hard. Algorithmically, however, the picture is different. Generic submodular maximization admits efficient algorithms that can attain *approximate* optima with global guarantees; these algorithms are typically based on local search techniques [16, 35]. In contrast, although polynomial-time solvable, submodular function minimization (SFM) which seeks to solve

$$\min_{S \subseteq V} F(S), \tag{1}$$

poses substantial algorithmic difficulties. This is partly due to the fact that one is commonly interested in an exact solution (or an arbitrarily close approximation thereof), and "polynomial-time" is not necessarily equivalent to "practically fast".

Submodular minimization algorithms may be obtained from two main perspectives: *combinatorial* and *continuous*. Combinatorial algorithms for SFM typically use close connections to matroid and

maximum flow methods; the currently theoretically fastest combinatorial algorithm for SFM scales as $O(n^6 + n^5 \tau)$, where $\tau$ is the time to evaluate the function oracle [37] (for an overview of other algorithms, see e.g., [33]). These combinatorial algorithms are typically nontrivial to implement.

Continuous methods offer an alternative by instead minimizing a *convex extension*. This idea exploits the fundamental connection between a submodular function $F$ and its *Lovász extension* $f$ [32], which is continuous and convex. The SFM problem (1) is then equivalent to

$$\min_{x \in [0,1]^n} f(x). \tag{2}$$

The Lovász extension $f$ is nonsmooth, so we might have to resort to subgradient methods. While a fundamental result of Edmonds [15] demonstrates that a subgradient of $f$ can be computed in $O(n \log n)$ time, subgradient methods can be sensitive to choices of the step size, and can be slow. They theoreticaly converge at a rate of $O(1/\sqrt{t})$ (after $t$ iterations). The "smoothing technique" of [36] does not in general apply here because computing a smoothed gradient is equivalent to solving the submodular minimization problem. We discuss this issue further in Section 2.

An alternative to minimizing the Lovász extension directly on $[0,1]^n$ is to consider a slightly modified convex problem. Specifically, the exact solution of the discrete problem $\min_{S \subseteq V} F(S)$ and of its nonsmooth convex relaxation $\min_{x \in [0,1]^n} f(x)$ may be found as a level set $S_0 = \{k \mid x_k^* \geqslant 0\}$ of the unique point $x^*$ that minimizes the strongly convex function [1, 10]:

$$f(x) + \tfrac{1}{2}\|x\|^2. \tag{3}$$

We will refer to the minimization of (3) as the *proximal* problem due to its close similarity to proximity operators used in convex optimization [12]. When $F$ is a cut function, (3) becomes a total variation problem (see, e.g., [9] and references therein) that also occurs in other regularization problems [1]. Two noteworthy points about (3) are: (i) addition of the strongly convex component $\tfrac{1}{2}\|x\|^2$; (ii) the ensuing removal of the *box-constraints* $x \in [0,1]^n$. These changes allow us to consider a convex dual which is amenable to smooth optimization techniques.

Typical approaches to generic SFM include Frank-Wolfe methods [17] that have cheap iterations and $O(1/t)$ convergence, but can be quite slow in practice (Section 5); or the minimum-norm-point/Fujishige-Wolfe algorithm [20] that has expensive iterations but finite convergence. Other recent methods are approximate [24]. In contrast to several iterative methods based on convex relaxations, we seek to obtain exact discrete solutions.

To the best of our knowledge, all generic algorithms that use only submodularity are several orders of magnitude slower than specialized algorithms when they exist (e.g., for graph cuts). However, the submodular function is not always generic and given via a black-box, but has known structure. Following [28, 29, 38, 41], we make the assumption that $F(S) = \sum_{i=1}^r F_i(S)$ is a sum of sufficiently "simple" functions (see Sec. 3). This structure allows the use of (parallelizable) dual decomposition techniques for the problem in Eq. (2), with [11, 38] or without [29] Nesterov's smoothing technique, or with direct smoothing [41] techniques. But existing approaches typically have two drawbacks: (1) they use smoothing or step-size parameters whose selection may be critical and quite tedious; and (2) they still exhibit slow convergence (see Section 5).

These drawbacks arise from working with formulation (2). Our main insight is that, despite seemingly counter-intuitive, the proximal problem (3) offers a much more user-friendly tool for solving (1) than its natural convex counterpart (2), both in implementation and running time. We approach problem (3) via its dual. This allows decomposition techniques which combine well with orthogonal projection and reflection methods that (a) exhibit faster convergence, (b) are easily parallelizable, (c) require no extra hyperparameters, and (d) are extremely easy to implement.

The main three algorithms that we consider are: (i) dual block-coordinate descent (equivalently, primal-dual proximal-Dykstra), which was already shown to be extremely efficient for total variation problems [2] that are special cases of Problem (3); (ii) Douglas-Rachford splitting using the careful variant of [4], which for our formulation (Section 4.2) requires no hyper-parameters; and (iii) accelerated projected gradient [5]. We will see these alternative algorithms can offer speedups beyond known efficiencies. Our observations have two implications: first, from the viewpoint of solving Problem (3), they offers speedups for often occurring denoising and reconstruction problems that employ total variation. Second, our experiments suggest that projection and reflection methods can work very well for solving the combinatorial problem (1).

In summary, we make the following contributions: (1) In Section 3, we cast the problem of minimizing decomposable submodular functions as an orthogonal projection problem and show how existing optimization techniques may be brought to bear on this problem, to obtain fast, easy-to-code and easily parallelizable algorithms. In addition, we show examples of classes of functions amenable to our approach. In particular, for *simple* functions, i.e., those for which minimizing $F(S) - a(S)$ is easy for all vectors[1] $a \in \mathbb{R}^n$, the problem in Eq. (3) may be solved in $O(\log \frac{1}{\varepsilon})$ calls to such minimization routines, to reach a precision $\varepsilon$ (Section 2,3). (2) In Section 5, we demonstrate the empirical gains of using accelerated proximal methods, Douglas-Rachford and block coordinate descent methods over existing approaches: fewer hyperparameters and faster convergence.

## 2 Review of relevant results from submodular analysis

The relevant concepts we review here are the Lovász extension, base polytopes of submodular functions, and relationships between proximal and discrete problems. For more details, see [1, 19].

**Lovász extension and convexity.** The power set $2^V$ may be naturally identified with the vertices of the hypercube, i.e., $\{0,1\}^n$. The Lovász extension $f$ of any set function is defined by linear interpolation, so that for any $S \subset V$, $F(S) = f(1_S)$. It may be computed in closed form once the components of $x$ are sorted: if $x_{\sigma(1)} \geqslant \cdots \geqslant x_{\sigma(n)}$, then $f(x) = \sum_{k=1}^n x_{\sigma(k)} \big[ F(\{\sigma(1), \ldots, \sigma(k)\}) - F(\{\sigma(1), \ldots, \sigma(k-1)\}) \big]$ [32]. For the graph cut function, $f$ is the total variation.

In this paper, we are going to use two important results: (a) if the set function $F$ is submodular, then its Lovász extension $f$ is convex, and (b) minimizing the set function $F$ is equivalent to minimizing $f(x)$ with respect to $x \in [0,1]^n$. Given $x \in [0,1]^n$, all of its level sets may be considered and the function may be evaluated (at most $n$ times) to obtain a set $S$. Moreover, for a submodular function, the Lovász extension happens to be the support function of the base polytope $B(F)$ defined as

$$B(F) = \{ y \in \mathbb{R}^n \mid \forall S \subset V, \ y(S) \leqslant F(S) \text{ and } y(V) = F(V) \},$$

that is $f(x) = \max_{y \in B(F)} y^\top x$ [15]. A maximizer of $y^\top x$ (and hence the value of $f(x)$), may be computed by the "greedy algorithm", which first sorts the components of $w$ in decreasing order $x_{\sigma(1)} \geqslant \cdots \geqslant x_{\sigma(n)}$, and then compute $y_{\sigma(k)} = F(\{\sigma(1), \ldots, \sigma(k)\}) - F(\{\sigma(1), \ldots, \sigma(k-1)\})$. In other words, a linear function can be maximized over $B(F)$ in time $O(n \log n + n\tau)$ (note that the term $n\tau$ may be improved in many special cases). This is crucial for exploiting convex duality.

**Dual of discrete problem.** We may derive a dual problem to the discrete problem in Eq. (1) and the convex nonsmooth problem in Eq. (2), as follows:

$$\min_{S \subseteq V} F(S) = \min_{x \in [0,1]^n} f(x) = \min_{x \in [0,1]^n} \max_{y \in B(F)} y^\top x = \max_{y \in B(F)} \min_{x \in [0,1]^n} y^\top x = \max_{y \in B(F)} (y)_-(V), \quad (4)$$

where $(y)_- = \min\{y, 0\}$ applied elementwise. This allows to obtain dual certificates of optimality from any $y \in B(F)$ and $x \in [0,1]^n$.

**Proximal problem.** The optimization problem (3), i.e., $\min_{x \in \mathbb{R}^n} f(x) + \frac{1}{2}\|x\|^2$, has intricate relations to the SFM problem [10]. Given the unique optimal solution $x^*$ of (3), the maximal (resp. minimal) optimizer of the SFM problem is the set $S^*$ of nonnegative (resp. positive) elements of $x^*$. More precisely, solving (3) is equivalent to minimizing $F(S) + \mu|S|$ for all $\mu \in \mathbb{R}$. A solution $S_\mu \subseteq V$ is obtained from a solution $x^*$ as $S_\mu^* = \{i \mid x_i^* \geqslant \mu\}$. Conversely, $x^*$ may be obtained from all $S_\mu^*$ as $x_k^* = \sup\{\mu \in \mathbb{R} \mid k \in S_\mu^*\}$ for all $k \in V$. Moreover, if $x$ is an $\varepsilon$-optimal solution of Eq. (3), then we may construct $\sqrt{\varepsilon n}$-optimal solutions for all $S_\mu$ [1; Prop. 10.5]. In practice, the duality gap of the discrete problem is usually much lower than that of the proximal version of the same problem, as we will see in Section 5. Note that the problem in Eq. (3) provides much more information than Eq. (2), as all $\mu$-parameterized discrete problems are solved.

The dual problem of Problem (3) reads as follows:

$$\min_{x \in \mathbb{R}^n} f(x) + \tfrac{1}{2}\|x\|_2^2 = \min_{x \in \mathbb{R}^n} \max_{y \in B(F)} y^\top x + \tfrac{1}{2}\|x\|_2^2 = \max_{y \in B(F)} \min_{x \in \mathbb{R}^n} y^\top x + \tfrac{1}{2}\|x\|_2^2 = \max_{y \in B(F)} -\tfrac{1}{2}\|y\|_2^2,$$

where primal and dual variables are linked as $x = -y$. Observe that this dual problem is equivalent to finding the orthogonal projection of $0$ onto $B(F)$.

**Divide-and-conquer strategies for the proximal problems.** Given a solution $x^*$ of the proximal problem, we have seen how to get $S^*_\mu$ for any $\mu$ by simply thresholding $x^*$ at $\mu$. Conversely, one can recover $x^*$ exactly from at most $n$ well-chosen values of $\mu$. A known divide-and-conquer strategy [19, 21] hinges upon the fact that for any $\mu$, one can easily see which components of $x^*$ are greater or smaller than $\mu$ by computing $S^*_\mu$. The resulting algorithm makes $O(n)$ calls to the submodular function oracle. In [25], we extend an alternative approach by Tarjan et al. [42] for cuts to general submodular functions and obtain a solution to (3) up to precision $\varepsilon$ in $O(\min\{n, \log \frac{1}{\varepsilon}\})$ iterations. This result is particularly useful if our function $F$ is a sum of functions for each of which by itself the SFM problem is easy. Beyond squared $\ell_2$-norms, our algorithm equally applies to computing all minimizers of $f(x) + \sum_{j=1}^p h_j(x_j)$ for arbitrary smooth strictly convex functions $h_j$, $j = 1, \ldots, n$.

## 3 Decomposition of submodular functions

Following [28, 29, 38, 41], we assume that our function $F$ may be decomposed as the sum $F(S) = \sum_{j=1}^r F_j(S)$ of $r$ "simple" functions. In this paper, by "simple" we mean functions $G$ for which $G(S) - a(S)$ can be minimized efficiently for all vectors $a \in \mathbb{R}^n$ (more precisely, we require that $S \mapsto G(S \cup T) - a(S)$ can be minimized efficiently over all subsets of $V \setminus T$, for any $T \subseteq V$ and $a$). Efficiency may arise from the functional form of $G$, or from the fact that $G$ has small support. For such functions, Problems (1) and (3) become

$$\min_{S \subseteq V} \sum_{j=1}^r F_j(S) = \min_{x \in [0,1]^n} \sum_{j=1}^r f_j(x) \qquad \min_{x \in \mathbb{R}^n} \sum_{j=1}^r f_j(x) + \tfrac{1}{2}\|x\|_2^2. \qquad (5)$$

The key to the algorithms presented here is to be able to minimize $\frac{1}{2}\|x - z\|_2^2 + f_j(x)$, or equivalently, to orthogonally project $z$ onto $B(F_j)$: $\min \frac{1}{2}\|y - z\|_2^2$ subject to $y \in B(F_j)$.

We next sketch some examples of functions $F$ and their decompositions into simple functions $F_j$. As shown at the end of Section 2, projecting onto $B(F_j)$ is easy as soon as the corresponding submodular minimization problems are easy. Here we outline some cases for which specialized fast algorithms are known.

**Graph cuts.** A widely used class of submodular functions are graph cuts. Graphs may be decomposed into substructures such as trees, simple paths or single edges. Message passing algorithms apply to trees, while the proximal problem for paths is very efficiently solved by [2]. For single edges, it is solvable in closed form. Tree decompositions are common in graphical models, whereas path decompositions are frequently used for TV problems [2].

**Concave functions.** Another important class of submodular functions is that of concave functions of cardinality, i.e., $F_j(S) = h(|S|)$ for a concave function $h$. Problem (3) for such functions may be solved in $O(n \log n)$ time (see [18] and our appendix in [25]). Functions of this class have been used in [24, 27, 41]. Such functions also include covering functions [41].

**Hierarchical functions.** Here, the ground set corresponds to the leaves of a rooted, undirected tree. Each node has a weight, and the cost of a set of nodes $S \subseteq V$ is the sum of the weights of all nodes in the smallest subtree (including the root) that spans $S$. This class of functions too admits to solve the proximal problem in $O(n \log n)$ time [22, 23, 26].

**Small support.** Any general, potentially slower algorithm such as the minimum-norm-point algorithm can be applied if the support of each $F_j$ is only a small subset of the ground set.

### 3.1 Dual decomposition of the nonsmooth problem

We first review existing dual decomposition techniques for the nonsmooth problem (1). We always assume that $F = \sum_{j=1}^r F_j$, and define $\mathcal{H}^r := \prod_{j=1}^r \mathbb{R}^n \simeq \mathbb{R}^{n \times r}$. We follow [29] to derive a dual formulation (see appendix in [25]):

**Lemma 1.** *The dual of Problem* (1) *may be written in terms of variables* $\lambda_1, \ldots, \lambda_r \in \mathbb{R}^n$ *as*

$$\max \sum_{j=1}^r g_j(\lambda_j) \qquad s.t. \quad \lambda \in \left\{ (\lambda_1, \ldots, \lambda_r) \in \mathcal{H}^r \mid \sum_{j=1}^r \lambda_j = 0 \right\} \qquad (6)$$

*where* $g_j(\lambda_j) = \min_{S \subset V} F_j(S) - \lambda_j(S)$ *is a nonsmooth concave function.*

The dual is the maximization of a nonsmooth concave function over a convex set, onto which it is easy to project: the projection of a vector $y$ has $j$-th block equal to $y_j - \frac{1}{r} \sum_{k=1}^r y_k$. Moreover, in our setup, functions $g_j$ and their subgradients may be computed efficiently through SFM.

We consider several existing alternatives for the minimization of $f(x)$ on $x \in [0,1]^n$, most of which use Lemma 1. Computing subgradients for any $f_j$ means calling the greedy algorithm, which runs in time $O(n \log n)$. All of the following algorithms require the tuning of an appropriate step size.

**Primal subgradient descent (primal-sgd)**: Agnostic to any decomposition properties, we may apply a standard simple subgradient method to $f$. A subgradient of $f$ may be obtained from the subgradients of the components $f_j$. This algorithm converges at rate $O(1/\sqrt{t})$.

**Dual subgradient descent (dual-sgd)** [29]: Applying a subgradient method to the nonsmooth dual in Lemma 1 leads to a convergence rate of $O(1/\sqrt{t})$. Computing a subgradient requires minimizing the submodular functions $F_j$ individually. In simulations, following [29], we consider a step-size rule similar to Polyak's rule (dual-sgd-P) [6], as well as a decaying step-size (dual-sgd-F), and use discrete optimization for all $F_j$.

**Primal smoothing (primal-smooth)** [41]: The nonsmooth primal may be smoothed in several ways by smoothing the $f_j$ individually; one example is $\tilde{f}_j^\varepsilon(x_j) = \max_{y_j \in B(F_j)} y_j^\top x_j - \frac{\varepsilon}{2}\|y_j\|^2$. This leads to a function that is $(1/\varepsilon)$-smooth. Computing $\tilde{f}_j^\varepsilon$ means solving the proximal problem for $F_j$. The convergence rate is $O(1/t)$, but, apart from step size which may be set relatively easily, the smoothing constant $\varepsilon$ needs to be defined.

**Dual smoothing (dual-smooth)**: Instead of the primal, the dual (6) may be smoothed, e.g., by entropy [8, 38] applied to each $g_j$ as $\tilde{g}_j^\varepsilon(\lambda_j) = \min_{x \in [0,1]^n} f_j(x) + \varepsilon h(x)$ where $h(x)$ is a negative entropy. Again, the convergence rate is $O(1/t)$ but there are two free parameters (in particular the smoothing constant $\varepsilon$ which is hard to tune). This method too requires solving proximal problems for all $F_j$ in each iteration.

Dual smoothing with entropy also admits coordinate descent methods [34] that exploit the decomposition, but we do not compare to those here.

## 3.2 Dual decomposition methods for proximal problems

We may also consider Eq. (3) and first derive a dual problem using the same technique as in Section 3.1. Lemma 2 (proved in the appendix in [25]) formally presents our dual formulation as a best approximation problem. The primal variable can be recovered as $x = -\sum_j y_j$.

**Lemma 2.** *The dual of Eq. (3) may be written as the* best approximation problem

$$\min_{\lambda, y} \quad \|y - \lambda\|_2^2 \qquad s.t. \ \lambda \in \Big\{ (\lambda_1, \ldots, \lambda_r) \in \mathcal{H}^r \mid \sum_{j=1}^r \lambda_j = 0 \Big\}, \qquad y \in \prod_{j=1}^r B(F_j). \quad (7)$$

We can actually eliminate the $\lambda_j$ and obtain the simpler looking dual problem

$$\max_y -\frac{1}{2}\Big\| \sum_{j=1}^r y_j \Big\|_2^2 \qquad s.t. \quad y_j \in B(F_j), \ j \in \{1, \ldots, r\} \quad (8)$$

Such a dual was also used in [40]. In Section 5, we will see the effect of solving one of these duals or the other. For the simpler dual (8) the case $r = 2$ is of special interest; it reads

$$\max_{y_1 \in B(F_1), \, y_2 \in B(F_2)} -\frac{1}{2}\|y_1 + y_2\|_2^2 \quad \Longleftrightarrow \quad \min_{y_1 \in B(F_1), -y_2 \in -B(F_2)} \|y_1 - (-y_2)\|_2. \quad (9)$$

We write problem (9) in this suggestive form to highlight its key geometric structure: it is, like (7), a *best approximation problem*: i.e., the problem of finding the closest point between the polytopes $B(F_1)$ and $-B(F_2)$. Notice, however, that (7) is very different from (9)—the former operates in a product space while the latter does not, a difference that can have impact in practice (see Section 5). We are now ready to present algorithms that exploit our dual formulations.

## 4 Algorithms

We describe a few competing methods for solving our smooth dual formulations. We describe the details for the special 2-block case (9); the same arguments apply to the block dual from Lemma 2.

### 4.1 Block coordinate descent or proximal-Dykstra

Perhaps the simplest approach to solving (9) (viewed as a minimization problem) is to use a block coordinate descent (BCD) procedure, which in this case performs the alternating projections:

$$y_1^{k+1} \leftarrow \operatorname{argmin}_{y_1 \in B(F_1)} \|y_1 - (-y_2^k)\|_2^2; \qquad y_2^{k+1} \leftarrow \operatorname{argmin}_{y_2 \in B(F_2)} \|y_2 - (-y_1^{k+1})\|_2. \quad (10)$$

The iterations for solving (8) are analogous. This BCD method (applied to (9)) is equivalent to applying the so-called proximal-Dykstra method [12] to the primal problem. This may be seen by comparing the iterates. Notice that the BCD iteration (10) is nothing but alternating projections onto the convex polyhedra $B(F_1)$ and $B(F_2)$. There exists a large body of literature studying method of alternating projections—we refer the interested reader to the monograph [13] for further details.

However, despite its attractive simplicity, it is known that BCD (in its alternating projections form), can converge arbitrarily slowly [4] depending on the relative orientation of the convex sets onto which one projects. Thus, we turn to a potentially more effective method.

## 4.2 Douglas-Rachford splitting

The Douglas-Rachford (DR) splitting method [14] includes algorithms like ADMM as a special case [12]. It avoids the slowdowns alluded to above by replacing alternating projections with alternating "reflections". Formally, DR applies to convex problems of the form [3, 12]

$$\min_x \quad \phi_1(x) + \phi_2(x), \tag{11}$$

subject to the qualification $\operatorname{ri}(\operatorname{dom}\phi_1) \cap \operatorname{ri}(\operatorname{dom}\phi_2) \neq \varnothing$. To solve (11), DR starts with some $z_0$, and performs the three-step iteration (for $k \geq 0$):

$$1.\ x_k = \operatorname{prox}_{\phi_2}(z_k); \qquad 2.\ v_k = \operatorname{prox}_{\phi_1}(2x_k - z_k); \qquad 3.\ z_{k+1} = z_k + \gamma_k(v_k - z_k), \tag{12}$$

where $\gamma_k \in [0, 2]$ is a sequence of scalars that satisfy $\sum_k \gamma_k(2 - \gamma_k) = \infty$. The sequence $\{x_k\}$ produced by iteration (12) can be shown to converge to a solution of (11) [3; Thm. 25.6].

Introducing the *reflection operator*

$$R_\phi := 2\operatorname{prox}_\phi - \mathrm{I},$$

and setting $\gamma_k = 1$, the DR iteration (12) may be written in a more symmetric form as

$$x_k = \operatorname{prox}_{\phi_2}(z_k), \qquad z_{k+1} = \tfrac{1}{2}[R_{\phi_1}R_{\phi_2} + \mathrm{I}]z_k, \quad k \geq 0. \tag{13}$$

Applying DR to the duals (7) or (9), requires first putting them in the form (11), either by introducing extra variables or by going back to the primal, which is unnecessary. This is where the special structure of our dual problem proves crucial, a recognition that is subtle yet remarkably important.

Instead of applying DR to (9), consider the closely related problem

$$\min_y \quad \delta_1(y) + \delta_2^-(y), \tag{14}$$

where $\delta_1$, $\delta_2^-$ are indicator functions for $B(F_1)$ and $-B(F_2)$, respectively. Applying DR directly to (14) does not work because usually $\operatorname{ri}(\operatorname{dom}\delta_1) \cap \operatorname{ri}(\operatorname{dom}\delta_2) = \varnothing$. Indeed, applying DR to (14) generates iterates that diverge to infinity [4; Thm. 3.13(ii)]. Fortunately, even though the DR iterates for (14) may diverge, Bauschke et al. [4] show how to extract convergent sequences from these iterates, which actually solve the corresponding best approximation problem; for us this is nothing but the dual (9) that we wanted to solve in the first place. Theorem 3, which is a simplified version of [4; Thm. 3.13], formalizes the above discussion.

**Theorem 3.** *[4] Let $\mathcal{A}$ and $\mathcal{B}$ be nonempty polyhedral convex sets. Let $\Pi_\mathcal{A}$ ($\Pi_\mathcal{B}$) denote orthogonal projection onto $\mathcal{A}$ ($\mathcal{B}$), and let $R_\mathcal{A} := 2\Pi_\mathcal{A} - \mathrm{I}$ (similarly $R_\mathcal{B}$) be the corresponding reflection operator. Let $\{z_k\}$ be the sequence generated by the DR method (13) applied to (14). If $\mathcal{A} \cap \mathcal{B} \neq \varnothing$, then $\{z_k\}_{k \geq 0}$ converges weakly to a fixed-point of the operator $T := \tfrac{1}{2}[R_\mathcal{A}R_\mathcal{B} + \mathrm{I}]$; otherwise $\|z_k\|_2 \to \infty$. The sequences $\{x_k\}$ and $\{\Pi_\mathcal{A}\Pi_\mathcal{B}z_k\}$ are bounded; the weak cluster points of either of the two sequences*

$$\{(\Pi_\mathcal{A}R_\mathcal{B}z_k, x_k)\}_{k \geq 0} \quad \{(\Pi_\mathcal{A}x_k, x_k)\}_{k \geq 0}, \tag{15}$$

*are solutions best approximation problem $\min_{a,b} \|a - b\|$ such that $a \in \mathcal{A}$ and $b \in \mathcal{B}$.*

The key consequence of Theorem 3 is that we can apply DR with impunity to (14), and extract from its iterates the optimal solution to problem (9) (from which recovering the primal is trivial). The most important feature of solving the dual (9) in this way is that absolutely no stepsize tuning is required, making the method very practical and user friendly.

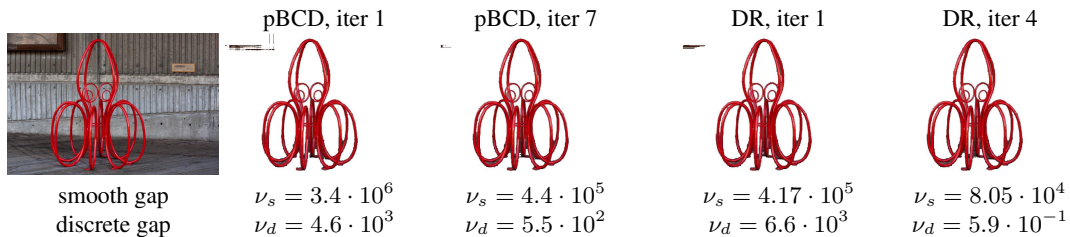

| | pBCD, iter 1 | pBCD, iter 7 | DR, iter 1 | DR, iter 4 |
|---|---|---|---|---|
| smooth gap | $\nu_s = 3.4 \cdot 10^6$ | $\nu_s = 4.4 \cdot 10^5$ | $\nu_s = 4.17 \cdot 10^5$ | $\nu_s = 8.05 \cdot 10^4$ |
| discrete gap | $\nu_d = 4.6 \cdot 10^3$ | $\nu_d = 5.5 \cdot 10^2$ | $\nu_d = 6.6 \cdot 10^3$ | $\nu_d = 5.9 \cdot 10^{-1}$ |

Figure 1: Segmentation results for the slowest and fastest projection method, with smooth ($\nu_s$) and discrete ($\nu_d$) duality gaps. Note how the background noise disappears only for small duality gaps.

## 5   Experiments

We empirically compare the proposed projection methods[2] to the (smoothed) subgradient methods discussed in Section 3.1. For solving the proximal problem, we apply block coordinate descent (BCD) and Douglas-Rachford (DR) to Problem (8) if applicable, and also to (7) (BCD-para, DR-para). In addition, we use acceleration to solve (8) or (9) [5]. The main iteration cost of all methods except for the primal subgradient method is the orthogonal projection onto polytopes $B(F_j)$. The primal subgradient method uses the greedy algorithm in each iteration, which runs in $O(n \log n)$. However, as we will see, its convergence is so slow to counteract any benefit that may arise from not using projections. We do not include Frank-Wolfe methods here, since FW is equivalent to a subgradient descent on the primal and converges correspondingly slowly.

As benchmark problems, we use (i) graph cut problems for segmentation, or MAP inference in a 4-neighborhood grid-structured MRF, and (ii) concave functions similar to [41], but together with graph cut functions. The functions in (i) decompose as sums over vertical and horizontal paths. All horizontal paths are independent and can be solved together in parallel, and similarly all vertical paths. The functions in (ii) are constructed by extracting regions $R_j$ via superpixels and, for each $R_j$, defining the function $F_j(S) = |S||R_j \setminus S|$. We use 200 and 500 regions. The problems have size $640 \times 427$. Hence, for (i) we have $r = 640 + 427$ (but solve it as $r = 2$) and for (ii) $r = 640 + 427 + 500$ (solved as $r = 3$). More details and experimental results may be found in [25].

**Two functions** ($r = 2$). Figure 2 shows the duality gaps for the discrete and smooth (where applicable) problems for two instances of segmentation problems. The algorithms working with the proximal problems are much faster than the ones directly solving the nonsmooth problem. In particular DR converges extremely fast, faster even than BCD which is known to be a state-of-the-art algorithms for this problem [2]. This, in itself, is a new insight for solving TV. If we aim for parallel methods, then again DR outperforms BCD. Figure 3 (right) shows the speedup gained from parallel processing. Using 8 cores, we obtain a 5-fold speed-up. We also see that the discrete gap shrinks faster than the smooth gap, i.e., the optimal discrete solution does not require to solve the smooth problem to extremely high accuracy. Figure 1 illustrates example results for different gaps.

**More functions** ($r > 2$). Figure 3 shows example results for four problems of sums of concave and cut functions. Here, we can only run DR-para. Overall, BCD, DR-para and the accelerated gradient method perform very well.

In summary, our experiments suggest that projection methods can be extremely useful for solving the combinatorial submodular minimization problem. Of the tested methods, DR, cyclic BCD and accelerated gradient perform very well. For parallelism, applying DR on (9) converges much faster than BCD on the same problem. Moreover, in terms of running times, running the DR method with a mixed Matlab/C implementation until convergence on a single core is only 3-8 times slower than the optimized efficient C code of [7], and only 2-4 times on 2 cores. These numbers should be read while considering that, unlike [7], the projection methods naturally lead to parallel implementations, and are able to integrate a large variety of functions.

## 6   Conclusion

We have presented a novel approach to submodular function minimization based on the equivalence with a best approximation problem. The use of reflection methods avoids any hyperparameters and reduce the number of iterations significantly, suggesting the suitability of reflection methods

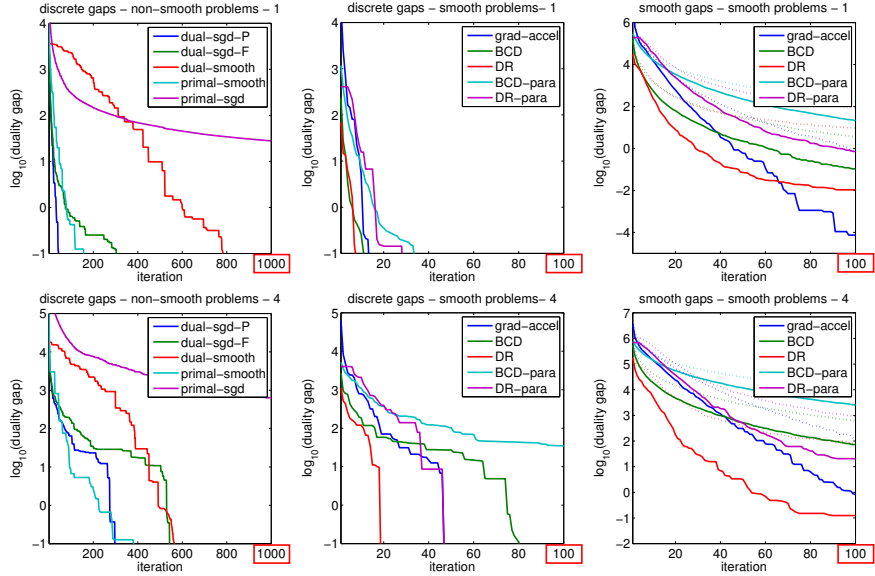

Figure 2: Comparison of convergence behaviors. Left: discrete duality gaps for various optimization schemes for the nonsmooth problem, from 1 to 1000 iterations. Middle: discrete duality gaps for various optimization schemes for the smooth problem, from 1 to 100 iterations. Right: corresponding continuous duality gaps. From top to bottom: two different images.

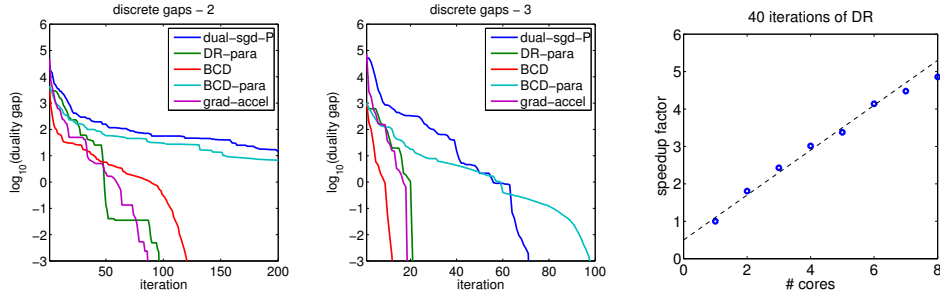

Figure 3: Left two plots: convergence behavior for graph cut plus concave functions. Right: Speedup due to parallel processing.

for combinatorial problems. Given the natural parallelization abilities of our approach, it would be interesting to perform detailed empirical comparisons with existing parallel implementations of graph cuts (e.g., [39]). Moreover, a generalization beyond submodular functions of the relationships between combinatorial optimization problems and convex problems would enable the application of our framework to other common situations such as multiple labels (see, e.g., [29]).

**Acknowledgments.** This research was in part funded by the Office of Naval Research under contract/grant number N00014-11-1-0688, by NSF CISE Expeditions award CCF-1139158, by DARPA XData Award FA8750-12-2-0331, and the European Research Council (SIERRA project), as well as gifts from Amazon Web Services, Google, SAP, Blue Goji, Cisco, Clearstory Data, Cloudera, Ericsson, Facebook, General Electric, Hortonworks, Intel, Microsoft, NetApp, Oracle, Samsung, Splunk, VMware and Yahoo!.

## Footnotes

[1] Every vector $a \in \mathbb{R}^n$ may be viewed as a modular (linear) set function: $a(S) \triangleq \sum_{i \in S} a(i)$.

[2]Code and data corresponding to this paper are available at *https://sites.google.com/site/mloptstat/drsubmod*

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
