[Reviews · NeurIPS 2013]

Submitted by Assigned_Reviewer_3

This paper casts the problem of minimizing decomposable submodular functions as an orthogonal projection problem to obtain easily parallelizable algorithms. Furthermore, by focusing on a special subclass of decomposable functions, they can prove the proximal problem can be solved in fewer iterations than currently possible.

The authors consider several existing methods for minimizing the Lovasz extension over [0,1]^n, experimentally verify their claims.

Unfortunately there is no empirical verification of the speedups in the case of implementation as a parallel algorithm, which is advertised as a selling point of the paper.

The paper is well-written and clear.
Summary: This paper casts the problem of minimizing decomposable submodular functions as an orthogonal projection problem to obtain easily parallelizable algorithms. Furthermore, by focusing on a special subclass of decomposable functions, they can prove the proximal problem can be solved in fewer iterations than currently possible.

Submitted by Assigned_Reviewer_5

* Summary

The paper presents an algorithm for exact submodular minimization based on solving a related strongly convex function via Douglas-Rachford splitting. A key insight in the paper is to construct a dual problem whose structure allows Douglas-Rachford splitting to be applied.

* Details

(Quality) The paper is technically strong but lacking in two main respects. First, the experimental evaluation while demonstrating fast convergence of the proposed method over competing approaches does not include sufficient detail for reproducibility. Moreover, the MRF problem can be solved by graphs cuts (for which online code is available). It would be interesting to compare running time of the proposed method against graph cuts for this problem (even with all the standard caveats---Matlab vs C, more general method vs specialized one, etc). I also do not understand why the parallel methods take more iterations to converge than the standard implementations in Figure 2. Can the authors please comment.

Second, while the paper demonstrates that the proposed method is empirically faster than competing approaches it does not provide any theoretical guarantees. Competing algorithms are shown to be O(1/t) or O(1/sqrt{t}). What is the running time of the Douglas-Rachford approach?

(Clarity) Over all the paper is clear written and (relative to the subject matter) easy to follow. Two suggestions that could improve the readability are:
1. On line 072 the authors discuss a solution method via level sets. However, details of this are missing here and the method only becomes apparent in section 2 (L150). Some clarification around line 072 would help.
2. Define y(A) = \sum_{a \in A} y_a around line 134.

(Originality) The approach makes use of known techniques but their combination is original.

(Significance) Given the pervasiveness of submodular functions in machine learning a fast and general minimization algorithm is likely to have an impact. The only drawback of the current method is that specialized methods are needed for projecting onto B(F_i) for each subproblem.
Summary: The paper presents a principled approach to minimizing decomposeable submodular functions. The approach is generally well described but some additional details could help improve the paper.

Submitted by Assigned_Reviewer_6

Summary of ideas:
There is a previously known proximal problem, whose minimization gives at least as much information than minimizing the Lovasz extension.
When a submodular function is written as a sum of submodular functions, each relatively easy to solve, solving the dual for the proximal problem can be done in terms of projections, rather than posing non-smooth functions as previous methods do.

Evaluation:

The paper presents a refined way of dealing with submodular functions that are additively decomposable into simpler functions, by novel dual decompositions of a less well known use of the Lovasz extension.
These allow the decomposition to proceed without some of the complications arising due to non-smoothness in previous approaches. The empirical evaluation using examples from image processing shows promising results.

The paper is very clear in describing its predecessors; unfortunately, this seems to leave insufficient space to be more than sketchy about the novel techniques.
For example, one of the important technical contributions is to show how the DR variant of reference [5] is applicable to dual (9), which is specialized for a sum of 2 functions; but [5] focuses on two functions as well, and how exactly to apply these algorithms for r > 2 is not clear to me, despite such results being reported as useful in the experimental evaluation section under the name DR-para.
We certainly do not know iteration complexity depends of r.
In another example, any detailed description of the third main contribution is essentially deferred to the supplementary material (which formally states neither an algorithm nor a theorem).

Pros:
- Decomposable submodular functions arise in applications, and more appropriate ways to minimize them are useful.
- The novel duals might spur further work on algorithms for their solution, beyond the algorithms proposed here.
Cons:
- While iteration and computational complexity bounds are treated as limitations of previous results, the novel optimization problems and algorithms are not accompanied by corresponding iteration complexity results. While a full analysis can be deferred to future work, some understanding of how r affects complexity under the different algorithms is missing.
- The presentation of the algorithms (as opposed to duals) is not clear enough.

Detailed comments:
- 072 "the level set {k, x_k^* \ge 0} of ... x^*" is not very clear, is {k | x_k^* \ge 0} meant? Ref [2] refers to many different problems, a more specific reference for the fact being mentioned would be useful.
- 088 Should reference section 3 where you give some examples in which this decomposition holds, and an idea of what types of "simplicity" we might encounter for the summand functions would be nice.
- 116 The notation a(A) where a\in R^n and A\subseteq V does not seem to be introduced anywhere.
- 117 This is misleading: Section 2 punts to the Supplementary material, which also skimps on explicit detail.
- 138 computed the -> computed by the
- 156 The term "discrete gaps" are used here and elsewhere but not clearly defined. The duality gaps of F(A) and a particular dual problem? Similarly for smooth gaps.
- 170 "In the appendix.." An explicit algorithm should be accompanied by a lemma and proof giving its complexity. Neither is present, in the paper nor in the supplementary material.
- "problems have size 640x427. Hence ... have r = 640 + 247" either there is a typo (427 to 247), or the relationship is not clear enough. Also, it is worth explaining how and why you "solve as r=2/3".
- Figure 2:
- Inconsistencies of vertical and horizontal axes makes comparisons difficult.
- "From top to bottom: four different images" presumably 2 different images. A wider benchmark
- Figure 3:
- Does "non-smooth problems" not correspond to the previously existing duals? the legends suggest that the new duals are being used.
- Empirical evaluation in general: comparisons of convergence in terms of iteration counts are informative only when iteration costs are roughly comparable. Are they?
Summary: Interesting duals are accompanied by not so well presented algorithms.
The presentation should focus and give explicit detail on the novel aspects at the expense of some detail on existing methods.
Author Feedback

Author rebuttal: We thank all reviewers for their careful reading and comments. We address the comments separately below.

We would like to remark that one novelty of the paper is to approach the classical problem of submodular function minimization from a new viewpoint, which allows applying techniques to it that lead to algorithms with a number of plus points: efficiency, parallel implementations and no parameter tuning. To our knowledge, this is the first work combining these three advantages for a range of submodular functions that are widely used in applications.


-- Assigned_Reviewer_3 --

"no empirical verification of the speedups ... as a parallel algorithm"

Figure 3 (rightmost plot) shows the empirical speedup when running the algorithm on a multicore machine. Figure 3 in the supplement shows an additional speedup plot.


-- Assigned_Reviewer_5 --

"reproducibility"

We will add more details to the final version and make both the data and code publicly available. The "r=2" experiments were segmentation problems on two images with four different parameter sets (results shown in the main paper and supplement). The energy for the segmentation problem was a standard formulation (edge weights proportional to exp(-||x_i-x_j||^2 / sigma), where x_i are the RGB pixel values; plus a linear term from GMM log-likelihoods). The "regions" functions are set up as in [40]. The superpixels were created by "Turbopixels" (Levinshtein et al, code available online). The algorithms were implemented in Matlab and partially Mex.

"running time compared to graph cuts"

See line 375 in the paper. A somewhat more efficient implementation takes, for the experiments in the paper, about 3-8 times the time of the highly tuned maxflow implementation of [8] when run on a single core, and just 2-4 times the time when run on 2 cores. As you say, the DR algorithm is far more general than the specialized Maxflow code.

"parallel methods take more time to converge in Fig.2"

The parallel methods use a different scheme: they update all y_j in (8) in parallel (using Jacobi updates, i.e., for updating one y_j all y_k-values from the previous iteration are used), whereas sequential DR and BCD update y_1, and then y_2 (using the latest y_1) in (9) (a Gauss-Seidel approach). The slower convergence rate may be thus attributed to the difference between Jacobi and Gauss-Seidel style iterations.

"running time"

Typically DR/BCD converge at O(1/t) worst case rate, but much faster convergence (linear or in some cases, even finite) is often seen. A complete theoretical characterization of our scheme's complexity (beyond what one may expect based on DR/BCD) is however currently under investigation. The strong empirical performance shown in the paper is in fact a further motivation for studying this scheme.

"Clarity"

Thank you, we will add those explanations.


-- Assigned_Reviewer_6 --

We will clarify the algorithms and their properties as much as possible.

"how... to apply algorithms to r \ge 2"

The algorithm for r \ge 2 reduces the problem to the case r=2 by using Eqn. (7), which has the two variable vectors lambda and y. To this formulation, we can apply the DR algorithm described in Theorem 3, where A = H^r and B = \Prod_j B(F_j), respectively. It iterates the operator T and uses (15) to obtain the final dual variables. (Using DR directly on (7) or (9) is, as the paper explains, is impossible.) Going from dual to the primal is specified in line 247.
Eqn. (8) is also amenable to a BCD approach (Sec. 4.1) with Gauss Seidel (sequential) updates. We will clarify this in the final version.

"iteration complexity"

Typically DR/BCD converge at O(1/t) worst case rate, but much faster convergence (linear or in some cases, even finite) is often seen. A complete theoretical characterization of our scheme's complexity (beyond what one may expect based on DR/BCD) is however currently under investigation, due to its strong empirical performance. Indeed, the strong empirical performance shown in the paper is a motivation for further detailed study of this method.

The iteration complexity itself will be independent of r (because ultimately, the analysis of the 2-block case applies); r only affects the cost of a single iteration: one needs to do linearly more updates of y_j in eqn (7), but the single updates become much faster, and they can be done in parallel.

"detailed description of the third main contribution ... deferred to the supplementary material"

The end of Section 2 states the main idea. Explaining the entire algorithm in detail would make the paper too long, therefore we had to push details to the supplement. We will provide some more detail in the main paper and a formal algorithm and proof in the supplement.

"Detailed comments": we will address those in the final version. Some replies:
- 072: yes
- 'discrete gap': duality gap of the discrete problem (see eqn. (4))
'smooth gap': duality gap of the smooth, proximal problem (line 161)
- 427 vs. 247: this is a typo, should be 427. We then use one F_j for each row and column. All rows, however, are independent and can be solved in parallel; the same for all columns. Hence this is effectively r=2 (rows and columns). Similarly for r=3 (rows, columns, concave functions).
- Figure 3: yes, this is a typo.
- iteration costs: yes, they are comparable, as the projection onto B(F) is the main cost and needed by all methods except for one (see lines 343/344).